# Factors That Affect E-Learning Platforms after the Spread of COVID-19: Post Acceptance Study

**Rana Saeed Al-Maroof [1], Khadija Alhumaid [2], Iman Akour [3] and Said Salloum [4,5,*]**

1   English Language & Linguistics Department, Al Buraimi University College, Al Buraimi 512, Oman; rana@buc.edu.om
2   College of Education, Zayed University, Abu Dhabi 144534, United Arab Emirates; khadija.alhumaid@zu.ac.ae
3   Information Systems Department, College of Computing & Informatics, University of Sharjah, Sharjah 27272, United Arab Emirates; iakour@sharjah.ac.ae
4   School of Science, Engineering, and Environment, University of Salford, Manchester M5 4WT, UK
5   Machine Learning and NLP Research Group, University of Sharjah, Sharjah 27272, United Arab Emirates
*   Correspondence: ssalloum@sharjah.ac.ae

**Abstract:** The fear of vaccines has led to population rejection due to various reasons. Students have had their own inquiries towards the effectiveness of the vaccination, which leads to vaccination hesitancy. Vaccination hesitancy can affect students' perception, hence, acceptance of e-learning platforms. Therefore, this research attempts to explore the post-acceptance of e-learning platforms based on a conceptual model that has various variables. Each variable contributes differently to the post-acceptance of the e-learning platform. The research investigates the moderating role of vaccination fear on the post-acceptance of e-learning platforms among students. Thus, the study aims at exploring students' perceptions about their post-acceptance of e-learning platforms where vaccination fear functions as a moderator. The current study depends on an online questionnaire that is composed of 29 items. The total number of respondents is 630. The collected data was implemented to test the study model and the proposed constructs and hypotheses depending on the Smart PLS Software. Fear of vaccination has a significant impact on the acceptance of e-learning platforms, and it is a strong mediator in the conceptual model. The findings indicate a positive effect of the fear of vaccination as a mediator in the variables: perceived ease of use and usefulness, perceived daily routine, perceived critical mass and perceived self-efficiency. The implication gives a deep insight to take effective steps in reducing the level of fear of vaccination, supporting the vaccination confidence among educators, teachers and students who will, in turn, affect the society as a whole.

**Keywords:** fear of vaccination; post-acceptance; critical mass; self-efficiency; daily routine

## 1. Introduction

Higher education institutions faced many challenges during the spread of COVID-19 even after the vaccination availability. The challenge necessitates a huge change in teaching and learning environments. During the spread of the pandemic, most schools and universities switched to virtual classrooms instead of the four-wall traditional classroom, which was the only alternative to implement the teaching strategies and goals [1–7]. University students have been shifted to a completely different environment where many challenges have to be met [8,9]. Understanding students' challenges can help in forming the required mechanism to assess students' understanding, achievement and success during the second wave of COVID-19 [10–15].

It is interestingly important to note that this challenge is deeply evident even after the vaccination availability. This stems from the fact that vaccination has been rejected by the population. Vaccination hesitancy is an influential factor that affects the world in general and the educational system in particular. Vaccine hesitancy is the reluctance to

various vaccines observed in many countries. This notion denotes that there is a kind of delay in acceptance or refusal of vaccines despite vaccine availability. This notion varies from one place to another and across time, and it has a close relation with convenience and confidence. In addition, it is affected by contextual factors, individual conception, group impact and people's trust in health services [16–23]. The vaccination hesitancy varies from one user to another depending on self-perception, workplace and perceived risk. Accordingly, vaccine hesitancy is the main challenge behind COVID-19 vaccine uptake; hence, the vaccine against COVID-19 will face many barriers in a post-crisis context [24–26]. Vaccination confidence stands in contrast with vaccination hesitancy. Vaccine confidence is considered as something that can be achieved "in itself." Confidence in vaccines is dependent on trust in the health care system in addition to trust in a socio-political context. The users' confidence in vaccination may not be stable because of the perceived risks connected with immunization. In fact, it may lead to lower vaccination coverage and immunity loss [27–29].

Based on the previous assumption, this study aims to explore the effect of vaccination hesitancy or fear on the post-acceptance of e-learning platforms where the challenge is still crucial and evident due to the vacancy rejection. To achieve the proposed aim, a conceptual framework was developed that contributes to the two elements under discussion, which are post-acceptance and fear of vaccination. Certain external variables are added to the conceptual model due to their direct relation with post-acceptance, namely, the perceived daily routine, the critical mass and self-efficiency [30–33]. The fear of vaccination functions as a moderator that can measure the relation between TAM theory, Flow theory, daily routine, critical mass and self-efficiency on one hand, post-acceptance of e-learning platforms on the other hand. Accordingly, the genuine contribution of the current study can be summarized as follows: First, the study investigates the effects of vaccination fear or hesitancy in the educational environment. This can be done by using an integrated research model to explore the effect of vaccination fear on the e-learning platforms. To put it differently, a conceptual model was developed that combines the TAM acceptance model [34] and Flow theory [35,36] to highlight the significance and predictableness of the results. Second, the current study intends to evaluate the effect of vaccination fear as a moderator within the conceptual framework constructs. Vaccination fear and hesitancy is a crucial element, and its consequences vary depending on country financial status, gender and age. Recent research has shown that vaccine hesitancy is higher in countries with low income among young women and older adults [37]. Third, this study has extended the model to include external variables that are closely related to the post-acceptance stage [38], which are daily routine and critical mass. To the best of our knowledge, this is the first attempt to investigate the post-acceptance of e-learning platforms depending on an integrated model where fear of vaccination is the mediator to fill a significant research gap in the relevant literature.

## 2. Literature Review

The literature review has revealed that previous studies have tackled the effect of COVID-19 on the different educational e-learning platforms. They include Zoom Microsoft Teams, Moodle, Google Classroom, virtual reality applications, etc. All the previously mentioned platforms were effective during the spread of the pandemic and provided a suitable solution to the challenge [39–41]. TAM has been the influential model in most of the previous. In fact, most of these studies focused on the two more influential constructs, which are the perceived ease of use and the perceived usefulness. The studies have adhered to the effective role of these two constructs in making students' adoption or acceptance is on-demand [40,42–44]. The extended model of TAM that UTAUT has also been used as a model to measure the effectiveness of the constructs during the pandemic. The studies in India depend on different technology acceptance models. However, the study by [39] extended their model by adding SUS that is crucial to explore the perceived usability. Following the same trend is research by [41], where the TAM model is extended by adding

certain external factors, including computer self-efficacy, innovativeness, computer anxiety, perceived enjoyment, social norm, content and system quality.

Due to the fact that the pandemic effect has been extended to include different parts of the world, the study varies in place. Some of the studies were found in China, Indonesia, Malaysia and Vietnam. All these studies have described the effect of e-learning platforms during the pandemic using surveys or online questionnaires among students in undergraduate educational institutions [42,45–47]. Similarly, studies in Europe and Romania focus on the usage of surveys or online questionnaires either to students or farmers. Their samples were different due to the differences in the aim of the study. Concerning the Europe area, the study aims at explaining the readiness of farmers towards new technology during the pandemic, whereas the latter seeks to explore the effect of the online platform on a sample of students during the pandemic [41,48]. In terms of India, different researchers have investigated the effect of COVID 19 on the educational environment in different cities. They have reached the conclusion that the e-learning platform was effective at keeping direct and indirect means of communication among different participants within the educational institutions [39,40].

It can be noticed clearly from the studies in the table that most of the studies are within the educational institutions where e-learning platforms have been dominating the process of teaching and learning. It is a way that guarantees that the change from the traditional class to the e-learning environment is safe and effective [2,3]. By the end, all these in these educational institutions can achieve their goals and objectives [44,45].

## 3. The Conceptual Model and Hypotheses

### 3.1. Perceived Daily Routine

The daily routine notion refers to the extent where technology can become part of normal work and the integration of technology into users' normal work routines. The utilization of technology in such a way that it becomes a part of the daily pattern and it is perceived as being normal elements in the users' life is what defines the daily routine use of technology [30,49,50]. The daily routine is an influential factor that is part of the post-acceptance model. The daily routine is affected by the effectiveness and utilization of outcomes. This implies that users of the technology will consider it as part of their daily routine if it enhances their extrinsic motivation. It can also facilitate the integration of technology and work processes [30]. However, it seems that the effect of the daily routine is variant from one user to another. The variation lies in the fact that users may have different work circumstances and different conceptions towards the integration of technology in their daily work [51]. Accordingly, it is hypothesized that:

**Hypotheses 1 (H1).** *Daily routine has a positive effect on the post-acceptance of e-learning platforms.*

### 3.2. Self-Efficiency

The concept of self-efficiency was firstly tackled by Albert Bandura, who proposed this concept as part of the social cognitive theory and as an influential prerequisite for effective learning behaviour. Self-efficiency explores users' perception of their ability to do different tasks and to finish the task properly [52,53]. The e-learning system is closely related to the users' self-efficiency in the classroom. The actual teaching practices inside classrooms are affected by teachers' ability to use technology effectively and significantly. Thus, if teachers have a high sense of efficiency, they will definitely do the required task properly. To be able to achieve that goal, we need to engage students in different activities, which will, in turn, encourage teachers to use these technologies more constantly and gradually to develop their competence [54–57].

During COVID-19, the pandemic circumstances have proven to affect self-efficiency in the educational environment. Many researchers proposed that self-efficiency has influentially been affected by COVID-19. Ref. [58] proposed that within the time of crisis (the pandemic), self-efficiency has affected work commitment among teachers. Similarly, Hernández-Padilla et al. (2020) asserted self-efficiency could affect the adoption of technology

during COVID-19 by claiming that the adoption can be stopped or even prevented when there is a consciousness of the bad consequences of the pandemic. Thus, self-efficiency can have a protective role during the pandemic as it may create a more flexible atmosphere that encourages the adoption of technology. The vaccination appears as a solution to stop the pandemic spread, yet many people reject it due to different reasons. One of the influential reasons to reject a vaccine is the distrust in the health system, which urges people to reject the vaccine [59–61]. Given the fact the fear of vaccination has affected users' mental and physical health, the present study hypothesized that:

**Hypotheses 2 (H2).** *Self-efficiency has a positive impact on the post-acceptance of the e-learning platform.*

### 3.3. TAM Theory

Fred Davis developed the 'Technology Acceptance Model (TAM), where he contributed to the concept of technology acceptance, adoption and post-acceptance. The constructs of this model, which are the perceived ease of use and the perceived usefulness have been considered as the conceptual factors that add to the post-acceptance of the technology. The perceived ease of use has to do with the effectiveness of the easiness factor on users' performance, whereas the perceived usefulness has to do with the concept of 'effort-free that enhances users' performance [34]. Based on that, it is hypothesized that:

**Hypotheses 3 (H3).** *The perceived ease of use has a positive impact on the post-acceptance of the e-learning platform.*

**Hypotheses 4 (H4).** *The perceived usefulness has a positive impact on the post-acceptance of the e-learning platform.*

### 3.4. Flow Theory

The flow theory was proposed by Csikszentmihalyi as a means to comprehend the users' motivation. Motivation is closely related to the psychological state where the cognitive feeling of efficiency and motivation controls the users [62,63]. The flow theory refers to the state where users are intensely involved in a particular activity. The experience of using the technology is so enjoyable that users will do it in any circumstance. Flow theory is connected with intrinsic motivation, specifically to self-motivation. Self-motivation is considered to be one of the best ways to learn, which can urge its users to do different activities with a high degree of inner joy.

Recent studies have proven flow may result in high effectiveness and a positive attitude. It may also lead to a high level of achievement of educational goals by motivating students to acquire certain goals [64–66]. During COVID-19, the same results were presented by different researchers. Students' motivation during COVID-19 affects their learning outcomes, success and satisfaction. The results signify that the motivation of students to study in an online environment during the COVID-19 pandemic is an important determinant of the learning outcome success and satisfaction [67,68]. The fear of vaccination, which is considered a tool to reduce the bad effect of the pandemic, has affected users' performance [69]. Accordingly, it is hypothesized that:

**Hypotheses 5 (H5).** *Perceived enjoyment has a positive effect on the post-acceptance of the e-learning platform.*

### 3.5. Critical Mass Theory

Critical mass theory indicates that a group of the population makes a huge contribution towards the adoption of certain actions. Therefore, the other individual thinks that the behaviour is significant and starts imitating the same behaviour. The influence of critical mass on technology adoption is crucial. Whenever a group of friends or users decide to use technology, the other group will do the same [70–72]. Thus, the hypothesis is as follows:

**Hypotheses 6 (H6).** *Critical mass positively affects the intention to e-learning platform post-acceptance.*

*3.6. Mediating Effect of Perceived Vaccination Fear*

COVID-19 Vaccine fear and hesitancy is on the rise among different population especially with the rise of the conspiracy theory of Coronavirus itself. This leads people to reject the vaccination; hence, the percentage of vaccination hesitancy is getting higher and higher [73]. The risk theory affects the acceptance of vaccination in general. Risk theory is related to certain cultural evaluations and danger regulations. They respond deeply toward risk based on their emotions that are themselves conditioned by cultural appraisals. In fact, individuals tend to evaluate risk information in a manner that encourages expected utility [74–77]. The perceived fear of vaccination is different across gender. Generally speaking, women's attitude towards vaccines is related to their negative experiences with health care institutions, whereas men's attitude towards the vaccine is concerned with their immune system. They believe that vaccine will weaken their immune system [78,79].

Concerning the COVID-19 vaccine, it seems that health literacy plays an influential role in vaccine rejection. Recent studies have shown that students may reject the vaccine due to their health literacy, especially among female students. They have a high degree of COVID-19 vaccine; therefore, they are willing to adopt health-protective behaviour. The perceived fear of the vaccine may lead to the spread of the COVID-19 infection [80–82]. Based on the previous discussion, it is hypothesized that the mediating effect of vaccination fear is as follows:

M1: Fear of vaccination mediates the effect of PRU on the post-acceptance of the e-learning platform.

M2: Fear of vaccination mediates the effect of PU on the post-acceptance of the e-learning platform.

M3: Fear of vaccination mediates the effect of PEOU on the post-acceptance of the e-learning platform.

M4: Fear of vaccination mediates the effect of SE on the post-acceptance of the e-learning platform.

M5: Fear of vaccination mediates the effect of PE on the post-acceptance of the e-learning platform.

M6: Fear of vaccination mediates the effect of PCM on the post-acceptance of the e-learning platform.

The proposed research models rely on these hypotheses, as shown in Figure 1.

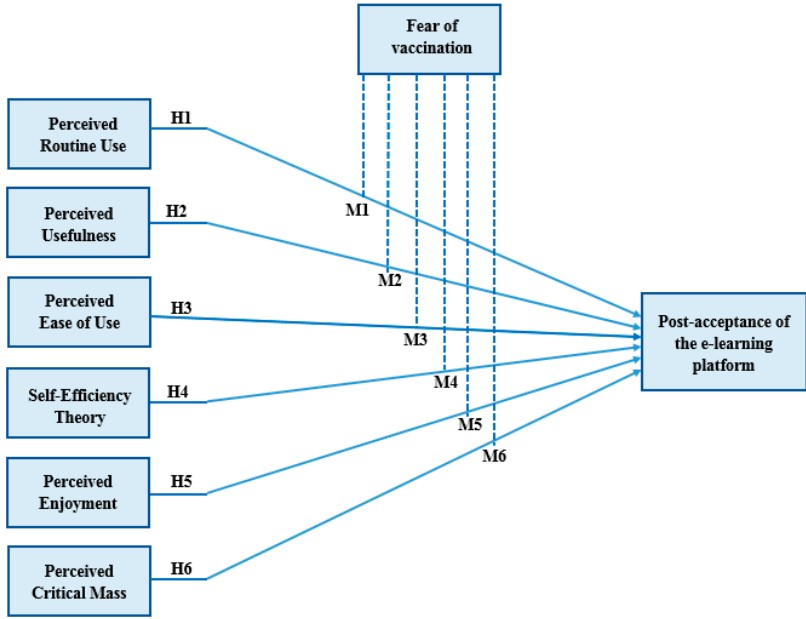

**Figure 1.** Research Model.

## 4. Research Methodology

### 4.1. Data Collection

Online surveys were distributed to the students studying in UAE universities to collect data in the course of the winter semester for 2020/2021. The data was collected during the period from 25 February 2021 to 5 May 2021. Students were handed 700 questionnaires on a random basis, of which 630 questionnaires were submitted by respondents. This amounted to a response rate of 90%. The remaining 70 questionnaires with missing values were rejected. In total, 630 questionnaires were effective to be used in the evaluation. Overall, 630 effective questionnaires with valid responses constituted a sufficient sample size for the study since [15] recommended a sampling size of 306 respondents for a population size equal to 1500. This shows that 630 effective questionnaires were quite more than the required 306 sample size and is deemed as an acceptable sample size for performing analysis with structural equation modelling [83] to test the hypotheses. The main point of interest is that hypotheses had been developed on the basis of existing theories; however, they were modified to the perspective of M-learning. The measurement model was evaluated through Structural equation modelling (SEM) and consequently handled using the final path model.

### 4.2. Personal/Demographic Information

Table 1 shows the data obtained for the personal/demographic information of respondents. While 57% of respondents were female students, 43% were males. While 63% of respondents belonged to the age group 18–29, 37% were older than 29. Most of the respondents had a sound educational background, and most of them held university degrees. Bachelor's degree was held by 68%, master's degree by 18%, doctoral degree by 7% respondents and diplomas were held by the rest. For studies with voluntary respondents, [84] recommended using a "purposive sampling approach." Individuals of varying ages and studying in various colleges that had been registered for studying different levels of different programs formed the study sample. Demographic info of the respondents was measured through IBM SPSS Statistics ver. 23. Complete demographic data of the respondents is given in Table 2.

**Table 1.** Studies on E-learning Platforms during the Pandemic.

| Author(s) | Research Setting | Theory | Method | Size-Samples | E-Learning Platform | Study Type |
|---|---|---|---|---|---|---|
| [42] | Indonesia | Technology Acceptance Model (TAM) with facilitating condition as the external factor | Survey | (974)sport science education students | E-learning systems | Adoption |
| [39] | India | TAM & SUS | Survey | University Students | online learning platforms (Microsoft Teams) | Perceived Usability |
| [41] | Europe | TAM and computer self-efficacy, innovativeness, computer anxiety, perceived enjoyment, social norm, content and system quality | Questionnaire | EU farmers and agricultural entrepreneurs | virtual reality applications | Adoption |
| [43] | Iraq | TAM | Questionnaire survey | 242 educators participate | Moodle | Acceptance |
| [40] | India | TAM | Survey | 125 responses from Faculty Members | Zoom platform | Adoption |

**Table 1.** *Cont.*

| Author(s) | Research Setting | Theory | Method | Size-Samples | E-Learning Platform | Study Type |
|---|---|---|---|---|---|---|
| [48] | Romania | N/A | Questionnaire | 206 university students | Virtual learning (Microsoft Teams, Google Classroom or Zoom | N/A |
| [45] | Malaysia | TAM | A Survey Questionnaire | undergraduate accounting students | online learning | Acceptance |
| [44] | Not specified | TAM | Survey | College Students | E-learning System | Acceptance |
| [46] | Indonesia | N/A | Survey | 502-public university Students | Moodle-based e-learning platform | behavioural intention |
| [85] | China | Attention, Relevance, Confidence, and Satisfaction (ARCS) theory | Interviews | College Students | online learning platform | Adoption |
| [47] | Vietnam | TAM | A bilingual questionnaire in English and Vietnamese | 30 participants in educational institutions | E-learning System | Acceptance |
| [86] | India | e-learning quality (ELQ) | Questionnaire | 435 undergraduate and graduate management students (international and national) | On-line Classes | Acceptance |
| [87] | India | UTAUT | Questionnaire | 430 Under Graduate students at GLA University | e-learning classes | Adoption |

**Table 2.** Demographic data of the respondents.

| Criterion | Factor | Frequency | Percentage |
|---|---|---|---|
| Gender | Female | 360 | 57% |
| | Male | 270 | 43% |
| Age | Between 18 to 29 | 398 | 63% |
| | Between 30 to 39 | 109 | 17% |
| | Between 40 to 49 | 69 | 11% |
| | Between 50 to 59 | 54 | 9% |
| Education qualification | Diploma | 45 | 7% |
| | Bachelor | 426 | 68% |
| | Master | 113 | 18% |
| | Doctorate | 46 | 7% |

*4.3. Study Instrument*

The instrument developed and used in this study for hypothesis testing was none other than a survey. This survey comprised of 29 items that evaluated 4 constructs included in the questionnaire. The sources of each of the 8 constructs have been depicted in Table 3. The study was made more applicable by including questions from earlier studies after making the necessary modifications.

**Table 3.** Measurement Items.

| Constructs | Items | Instrument | Sources |
|---|---|---|---|
| Post-Acceptance of e-learning Technology | EPOS1 | My use of EP is continued even after COVID-19. | [88] |
| | EPOS2 | | |
| Perceived Ease of Use | PEOU1 | I think EP is easy for me. | [89,90] |
| | PEOU2 | I think attending my classes via EP is easy. | |
| | PEOU3 | I think being skilful at using EP is easy. | |
| | PEOU4 | Lack of COVID 19 fear makes my daily use of EP easy. | |
| Perceived Usefulness | PU1 | I find EP to be advantageous. | [89,90] |
| | PU2 | Using EP would improve my effectiveness in my daily classes | |
| | PU3 | Using EP is not time-consuming when I do my exams and assignments. | |
| | PU4 | Lack of COVID 19 fear makes my daily use of EP more useful. | |
| Perceived Routine Use | PRU1 | My use of EP is part of my regular class practices. | [30] |
| | PRU2 | My use of EP is integrated to be part of my study routine. | |
| | PRU3 | My use of EP is currently a normal part of my study. | |
| | PRU4 | Lack of COVID 19 fear improves my daily routine. | |
| Perceived Enjoyment | PE1 | I find using EP for studying is fun. | [35,36] |
| | PE2 | I find using EP for studying is pleasant. | |
| | PE3 | I find using EP for studying exciting. | |
| | PE4 | Lack of fear of COVID-19 makes my study more enjoyable. | |
| Perceived Critical Mass | PCM1 | Most of my classmates and teachers regularly use EP for studying. | [31,91] |
| | PCM2 | Most of the people I contact them use EP frequently for studying. | |
| | PCM3 | Most of my friends often use EP for studying. | |
| | PCM4 | Most of my classmates and teachers have no fear of COVID-19. | |
| Self-efficiency | SE1 | I would be able to use EP because I got good experience in using it. | [32,33] |
| | SE2 | I would be able to use EP because my teachers gave me clear directions. | |
| | SE3 | I would be able to use EP because I had been exposed to EP before. | |
| | SE4 | Lack of COVID-19 fear makes me more professional in using EP. | |
| Fear of Vaccination | POV1 | I am no longer afraid of COVID-19. | [92,93] |
| | POV2 | I am not afraid of COVID-19 when I use EP in my study. | |
| | POV3 | I believe that the effect of COVID-19 on my study becomes less. | |

*4.4. Pilot Study of the Questionnaire*

The questionnaire items were evaluated with respect to their reliability during a pilot study. In total, 70 students were selected out of the decided population on a random basis to take part in this study. The research standards were followed while determining the sample size, which was computed as 10% of the entire sample size of the research (i.e., 700). The pilot study outcomes were evaluated through Cronbach's alpha test with the help of SmartPls (version 3) that determined the internal reliability. The measurement items were found to be acceptable. Usually, the acceptable value of the reliability coefficient is 0.70 in research conducted in the domain of social science [94]. Values of Cronbach's alpha obtained for 7 measurement scales have been shown in Table 4.

*4.5. Survey Structure*

The researcher circulated the questionnaire survey: the students studying in the universities of United Arab Emirates (N = 700) were handed over the online surveys. The study involved three highly reputed UAE universities.

The prepared questionnaire survey was distributed among students [84]. The following sections were part of the survey.

Section 1 collected the personal data of the participants.

Section 2 included 26 items that obtained the participants' perspective about e-learning systems.

Section 3 had three items that dealt with the Lack of COVID-19 Fear.

29 items in the questionnaire were measured with the help of a five-point Likert Scale whereby the scores were allocated as follows: strongly agreed (5), agree (4), neutral (3), disagree (2) and strongly disagree (1).

**Table 4.** Cronbach's Alpha values for the pilot study (Cronbach's Alpha $\geq$ 0.70).

| Constructs | Cronbach's Alpha |
| --- | --- |
| EPOS | 0.803 |
| PEOU | 0.869 |
| PU | 0.842 |
| PRU | 0.806 |
| PE | 0.838 |
| PCM | 0.814 |
| SE | 0.853 |
| POV | 0.804 |

Note: EPOS, Post-Acceptance of E-learning Technology; PEOU, Perceived Ease of Use; PU, Perceived Usefulness; PRU, Perceived Routine Use; PE, Perceived Enjoyment; PCM, Perceived Critical Mass; SE, Self-efficiency; POV, Fear of Vaccination.

## 5. Findings and Discussion

### 5.1. Data Analysis

The data analysis for the current study was carried out with the help of the partial least squares-structural equation modelling (PLS-SEM) via SmartPLS V.3.2.7 software [95]. The analysis of collected data made use of a dual-stage methodology for assessment whereby the measurement model was used in one stage while the structural model in the other [96]. There are a number of factors behind the use of PLS-SEM in the current study. The first reason is that PLS-SEM is the most appropriate option for studies that are based on any existing theory [97]. Another reason is that PLS-SEM is also appropriate for exploratory research with complex models [98]. Thirdly, there is no model fragmentation in PLS-SEM, and the entire model is evaluated holistically [99]. The last reason is that PLS-SEM gives accurate estimations due to the possibility of measurement and structural model analysis at the same time [100].

### 5.2. Convergent Validity

As per the recommendations of [96], the measurement model will be evaluated by determining the construct reliability and validity where construct reliability is computed through Cronbach's alpha, McDonald's omega and composite reliability, while validity is computed through convergent and discriminant validity. Considering the measurement of construct reliability, Cronbach's alpha values were determined and were in the range of 0.707 to 0.877 as seen in Table 5; Cronbach's alpha values exceed the 0.7 threshold value [101]. Table 5 also depicts that the value of McDonald's omega within 0.789 and 0.871, which is quite higher than the suggested value of 0.7 [102,103]. On the other hand, the results in composite reliability (CR) showed values from 0.769 to 0.899, which also exceed the 0.7 threshold value as evident from Table 4 [104]. Hence, no error was found in any of the constructs as the values of CR and Cronbach's alpha are greater than the threshold; thus, confirming construct reliability.

The values of average variance extracted (AVE) and factor loading must be determined in order to measure convergent validity [96]. It is evident from Table 5 that each factor loading exceeded 0.7, which is the threshold value. Moreover, AVE values (from 0.563 to

0.715) were also found to be higher than the threshold value (0.5) according to Table 5; thus, indicating the approval of convergent validity for each and every construct.

**Table 5.** Convergent validity results that assure acceptable values (Factor loading, Cronbach's Alpha, McDonald's omega, composite reliability ≥ 0.70 and AVE > 0.5).

| Constructs | Items | Factor Loading | Cronbach's Alpha | McDonald's ω | CR | AVE |
|---|---|---|---|---|---|---|
| Post-Acceptance of E-learning Technology | EPOS1 | 0.777 | 0.770 | 0.789 | 0.869 | 0.630 |
| | EPOS2 | 0.840 | | | | |
| Perceived Routine Use | PRU1 | 0.849 | 0.877 | 0.871 | 0.876 | 0.563 |
| | PRU2 | 0.729 | | | | |
| | PRU3 | 0.742 | | | | |
| | PRU4 | 0.793 | | | | |
| Perceived Ease of Use | PEOU1 | 0.744 | 0.707 | 0.792 | 0.819 | 0.701 |
| | PEOU2 | 0.721 | | | | |
| | PEOU3 | 0.857 | | | | |
| | PEOU4 | 0.850 | | | | |
| Perceived Usefulness | PU1 | 0.828 | 0.850 | 0.849 | 0.895 | 0.715 |
| | PU2 | 0.803 | | | | |
| | PU3 | 0.891 | | | | |
| | PU4 | 0.865 | | | | |
| Perceived Enjoyment | PE1 | 0.732 | 0.797 | 0.793 | 0.851 | 0.629 |
| | PE2 | 0.810 | | | | |
| | PE3 | 0.896 | | | | |
| | PE4 | 0.835 | | | | |
| Perceived Critical Mass | PCM1 | 0.800 | 0.852 | 0.868 | 0.769 | 0.636 |
| | PCM2 | 0.845 | | | | |
| | PCM3 | 0.842 | | | | |
| | PCM4 | 0.746 | | | | |
| Self-efficiency | SE1 | 0.801 | 0.789 | 0.799 | 0.899 | 0.601 |
| | SE2 | 0.873 | | | | |
| | SE3 | 0.814 | | | | |
| | SE4 | 0.822 | | | | |
| Fear of Vaccination | POV1 | 0.796 | 0.870 | 0.853 | 0.868 | 0.645 |
| | POV2 | 0.869 | | | | |
| | POV3 | 0.834 | | | | |

### 5.3. Discriminant Validity

The two criteria, namely, the Fornell–Larker criterion and the Heterotrait–Monotrait ratio (HTMT), had to be determined in order to measure the discriminant validity [96]. As is evident from Table 6, the Fornell–Larker criterion was fulfilled since the square root of each value of AVE exceeds the value of correlation with the rest of the constructs [105].

The values of the HTMT ratio have been shown in Table 7. These values imply that the constructs show HTMT values less than the 0.85 threshold value [106]; hence, confirming the HTMT ratio. Accordingly, the same data confirms the discriminant validity. The hassle-free assessment of reliability and validity of the measurement model reveals the possibility of using the collected data for the assessment of the structural model also.

### 5.4. Model Fit

The fit measures, namely, the standard root mean square residual (SRMR), exact fit criteria, d_ULS, d_G, Chi-Square, NFI and RMS_theta, are presented by SmartPLS, which represent the fitting of the model with respect to PLS-SEM [98]. The variation found between correlations based on observed values and correlation matrix based on the model is indicated by the SRMR [98]; the model fit measure is deemed to be good when the values of SRMR are below 0.08 [107]. Similarly, a model fit is deemed good when NFI values exceed 0.90 [108]. The ratio of the proposed model's Chi2 value with the null or benchmark model's Chi2 value gives the NFI [98]. NFI shows greater values for large parameters, which makes it a bad indicator of model fitness [98]. The discrepancy between the empirical covariance matrix and covariance matrix derived from the composite factor model is given by metrics, namely, the geodesic distance (d_G) and the squared Eucledian distance (d_ULS) [98,109]. Only in the case of reflective models, it is possible to apply RMS theta for determining the degree of outer model residuals correlation [110]. The quality of the PLS-SEM model increases as the value of RMS theta approaches zero; RMS values below 0.12 indicate good model fit while other values indicate the absence of fit [98,111]. According to [98], the correlation between the constructs is determined through the saturated model, while the overall impact and model structure is reflected by the estimated model.

**Table 6.** The Fornell–Larcker Scale.

|  | **EPOS** | **PEOU** | **PU** | **PRU** | **PE** | **PCM** | **SE** | **POV** |
|---|---|---|---|---|---|---|---|---|
| EPOS | **0.883** |  |  |  |  |  |  |  |
| PEOU | 0.474 | **0.809** |  |  |  |  |  |  |
| PU | 0.491 | 0.522 | **0.830** |  |  |  |  |  |
| PRU | 0.465 | 0.544 | 0.654 | **0.889** |  |  |  |  |
| PE | 0.593 | 0.404 | 0.532 | 0.534 | **0.847** |  |  |  |
| PCM | 0.568 | 0.679 | 0.508 | 0.477 | 0.427 | **0.823** |  |  |
| SE | 0.539 | 0.531 | 0.560 | 0.502 | 0.447 | 0.593 | **0.872** |  |
| POV | 0.475 | 0.509 | 0.589 | 0.554 | 0.541 | 0.362 | 0.503 | **0.848** |

Note: EPOS, Post-Acceptance of E-learning Technology; PEOU, Perceived Ease of Use; PU, Perceived Usefulness; PRU, Perceived Routine Use; PE, Perceived Enjoyment; PCM, Perceived Critical Mass; SE, Self-efficiency; POV, Fear of Vaccination. The square root of the AVE scores are shown in the bold diagonal constituents of the table, while the correlations between the constructs are shown by the off load diagonal constituents.

**Table 7.** The Heterotrait–Monotrait Ratio (HTMT).

|  | **EPOS** | **PEOU** | **PU** | **PRU** | **PE** | **PCM** | **SE** | **POV** |
|---|---|---|---|---|---|---|---|---|
| EPOS |  |  |  |  |  |  |  |  |
| PEOU | 0.633 |  |  |  |  |  |  |  |
| PU | 0.500 | 0.405 |  |  |  |  |  |  |
| PRU | 0.648 | 0.561 | 0.482 |  |  |  |  |  |
| PE | 0.554 | 0.559 | 0.572 | 0.509 |  |  |  |  |
| PCM | 0.693 | 0.523 | 0.683 | 0.511 | 0.747 |  |  |  |
| SE | 0.599 | 0.430 | 0.634 | 0.533 | 0.627 | 0.682 |  |  |
| POV | 0.644 | 0.415 | 0.477 | 0.617 | 0.747 | 0.572 | 0.693 |  |

Note: EPOS, Post-Acceptance of E-learning Technology; PEOU, Perceived Ease of Use; PU, Perceived Usefulness; PRU, Perceived Routine Use; PE, Perceived Enjoyment; PCM, Perceived Critical Mass; SE, Self-efficiency; POV, Fear of Vaccination.

The RMS_theta depicted a value of 0.079 in Table 8. Hence, the validity of the global PLS model is proved due to significantly high goodness-of-fit for the PLS-SEM model.



**Table 8.** Model fit indicators.

| Criteria | Complete Model | |
|---|---|---|
| | **Saturated Model** | **Estimated Mod** |
| SRMR | 0.036 | 0.037 |
| d_ULS | 0.773 | 1.392 |
| d_G | 0.559 | 0.561 |
| Chi-Square | 479.155 | 479.290 |
| NFI | 0.817 | 0.817 |
| Rms Theta | 0.079 | |

*5.5. Hypotheses Testing Using PLS-SEM*

Subsequent to the confirmation of the measurement model, the study will evaluate the structural model [112,113]. For structural model analysis, 5000 resamples are subjected to bootstrapping to determine the value of path coefficients and coefficient of determination ($R^2$) [114,115]. All the values for path coefficients, *t*-values, and *p*-values pertaining to all hypotheses determined during path analysis have been shown in Table 9. The results indicate support for all hypotheses.

**Table 9.** Hypotheses-testing of the research model (significant at ** $p \leq 0.01$, * $p < 0.05$).

| H | Relationship | Path | *t*-Value | *p*-Value | Direction | Decision |
|---|---|---|---|---|---|---|
| H1 | PRU → EPOS | 0.559 | 18.332 | 0.000 | Positive | Supported ** |
| H2 | PU → EPOS | 0.770 | 19.619 | 0.000 | Positive | Supported ** |
| H3 | PEOU → EPOS | 0.562 | 10.421 | 0.000 | Positive | Supported ** |
| H4 | SE → EPOS | 0.309 | 4.287 | 0.024 | Positive | Supported * |
| H5 | PE → EPOS | 0.636 | 15.497 | 0.000 | Positive | Supported ** |
| H6 | PCM → EPOS | 0.465 | 17.282 | 0.000 | Positive | Supported ** |

Note: EPOS, Post-Acceptance of E-learning Technology; PEOU, Perceived Ease of Use; PU, Perceived Usefulness; PRU, Perceived Routine Use; PE, Perceived Enjoyment; PCM, Perceived Critical Mass; SE, Self-efficiency.

The value of the coefficient of determination ($R^2$) must be found in order to evaluate the structural model [116]. The main aim of this coefficient is to measure the model's predictive accuracy. The $R^2$ can be mathematically defined as the square of the correlation of the actual value of a particular endogenous construct with its predicted value [98,117]. The $R^2$ coefficient represents the cumulative impact of all exogenous latent variables over an endogenous latent variable. Besides this, the coefficient also indicates the square of the value of the correlation of variables' actual value with their predicted values; thus, indicating the extent of variance among the endogenous constructs. It was proposed by [118] that the value of the coefficient is considered as high if it exceeds 0.67. Moreover, direct values are those ranging from 0.33 to 0.67, while weak values are those between 0.19 and 0.33. Conversely, values less than 0.19 are inadmissible. The model's moderate power of prediction is evident from 63.2% variance in Post-Acceptance of E-learning Technology, as evident in Table 10 and Figure 2.

**Table 10.** $R^2$ of the endogenous latent variables.

| Constructs | $R^2$ | Results |
|---|---|---|
| EPOS | 0.632 | Moderate |

Note: EPOS, Post-Acceptance of E-learning Technology.

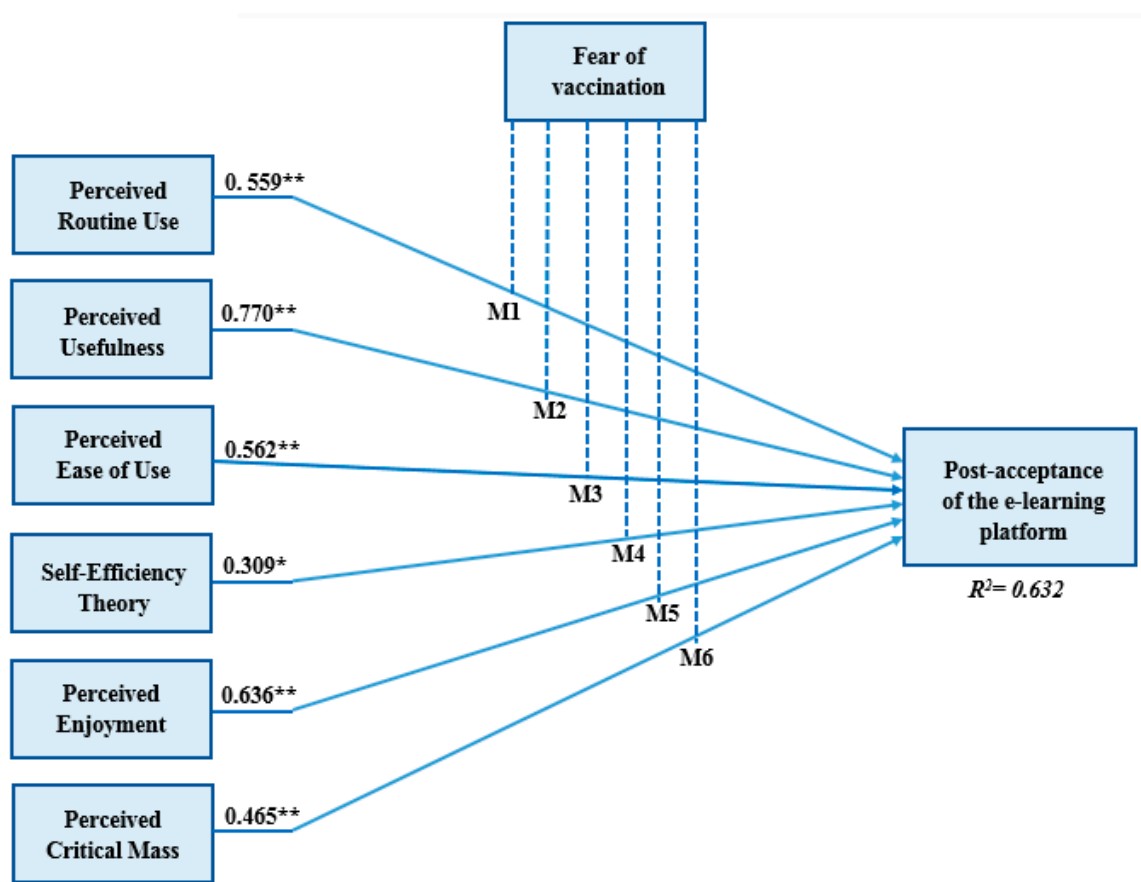

**Figure 2.** Path coefficient of the model (significant at ** $p \leq 0.01$, * $p < 0.05$).

Perceived Routine Use (PRTN), Perceived Usefulness (PU), Perceived Ease of Use (PEOU), Self-efficiency (EPSE), Perceived Enjoyment (PE) and Perceived Critical Mass (PCM) has significant effects on Post-Acceptance of E-learning Technology (EPOS) ($\beta = 0.559$, $p < 0.001$), ($\beta = 0.770$, $p < 0.001$), ($\beta = 0.562$, $p < 0.001$), ($\beta = 0.309$, $p < 0.05$), ($\beta = 0.636$, $p < 0.001$) and ($\beta = 0.465$, $p < 0.001$) respectively; hence, H1, H2, H3, H4, H5 and H6 are supported.

Moderating effect exercised by Fear of vaccination over apparent Perceived Ease of Use (PEOU), Perceived Usefulness (PU), Perceived Routine Use (PRTN), Perceived Enjoyment (PE), Perceived Critical Mass (PCM), and Self-efficiency (EPSE) of constructs was unfolded through additional testing. The effect of variables on direction or intensity of relationship among dependent and independent variables can be explained through the moderator effect. Table 11 depicts the outcomes of the current analysis, which show that all other hypotheses were accepted, signifying that perception of Perceived Ease of Use (PEOU), Perceived Usefulness (PU), Perceived Routine Use (PRTN), Perceived Enjoyment (PE), Perceived Critical Mass (PCM) and Self-efficiency (EPSE) of constructs' relationship is affected by Lack ofCOVID-19 Fear, whereby Lack of COVID-19 Fear was used as a moderator.

**Table 11.** Moderator Analysis Result.

| H | Relationship | Path a IV → Mediator | Path b Mediator → DV | Indirect Effect | SE Standard Deviation | *t*-Value | Bootstrapped Confidence Interval 95% LL | Bootstrapped Confidence Interval 95% UL | Decision |
|---|---|---|---|---|---|---|---|---|---|
| M1 | PRU * Fear of Vaccination → EPOS | 0.323 | 0.639 | 0.206 | 0.047 | 5.231 | 0.113 | 0.299 | Supported |
| M2 | PU * Fear of Vaccination → EPOS | 0.683 | 0.639 | 0.436 | 0.062 | 5.713 | 0.316 | 0.557 | Supported |

**Table 11.** *Cont.*

| H | Relationship | Path a IV → Mediator | Path b Mediator → DV | Indirect Effect | SE Standard Deviation | t-Value | Bootstrapped Confidence Interval | | Decision |
| | | | | | | | 95% LL | 95% UL | |
|---|---|---|---|---|---|---|---|---|---|
| M3 | PEOU * Fear of Vaccination → EPOS | 0.558 | 0.639 | 0.357 | 0.056 | 6.223 | 0.248 | 0.466 | Supported |
| M4 | SE * Fear of Vaccination → EPOS | 0.242 | 0.639 | 0.155 | 0.060 | 4.291 | 0.037 | 0.272 | Supported |
| M5 | PE * Fear of Vaccination → EPOS | 0.356 | 0.639 | 0.227 | 0.076 | 4.690 | 0.078 | 0.377 | Supported |
| M6 | PCM * Fear of Vaccination → EPOS | 0.648 | 0.639 | 0.414 | 0.059 | 7.014 | 0.299 | 0.529 | Supported |

Note: EPOS, Post-Acceptance of E-learning Technology; PEOU, Perceived Ease of Use; PU, Perceived Usefulness; PRU, Perceived Routine Use; PE, Perceived Enjoyment; PCM, Perceived Critical Mass; SE, Self-efficiency; POV, Fear of Vaccination.

## 6. Discussion and Conclusions

The present study has investigated the main variables that affect the acceptance of e-learning platform adapting TAM model with external factors of flow theory, perceived critical mass and perceived daily routine, moderated by fear of vaccination. Certain significant results will be discussed as follows. Firstly, it has been found out that all the variables are influential and are related to the independent variable. This result supports the TAM in the literature where the two constructs of perceived ease of use and perceived usefulness have been confirmed as having a positive effect on the acceptance of technology [14,41,119–124]. In a study by [125], the variable of perceived ease of use and perceived usefulness are considered as having a strong effect on the acceptance of Zoon as an e-learning platform, and they have a positive correlation with self-efficacy. The present study reveals that both PEOU and PU have a significant and positive impact on the post-acceptance of the e-learning platform.

Secondly, the hypotheses related to perceived daily routine and the perceived enjoyment have been supported as well. These findings are in agreement with the results from previous literature. In a study by [30], the daily routine has a positive impact on the acceptance of technology, and it has a direct relation with motivation. Similarly, the previous studies have shown that the perceived enjoyment affect significantly the acceptance of technology [41,126,127]. In a study by [128], it has been concluded that perceived enjoyment will positively affect the acceptance of technology and this variable has a close correlation with fast internet and well-built infrastructure.

Thirdly, critical mass and self-efficiency have been supported in the statistical analysis. The result proves that people try to be supportive in the time of crisis. They help each other to avoid any technical troubles. Thus, these two variables can have a protective role during a crisis [58,70,71,129]. In a study by [41], it has been shown that self-efficiency along with TAM (PEOU and PU) are the most influential factors that affect the acceptance of technology

Finally, the fear of vaccination or vaccination hesitancy mediates between significant variables. Based on the obtained findings, fear of vaccination is a strong mediator because it has a significant impact on the acceptance of e-learning platforms. The findings indicate a positive effect of the fear of vaccination on the main model-constructs. Previous studies have shown that vaccine hesitancy is a crucial challenge as it will create many barriers in a post-crisis context [24–26]. Therefore, students should have vaccination confidence which stands in contrast with vaccination hesitancy, and then they will be able to develop a kind of trust in the health care system to accept the vaccination and get rid of their fear. Accordingly, the fear of vaccination perfectly moderates the relationship between the proposed variables. Therefore, this study differentiates itself by adapting the fear of vaccination as a moderating effect on the post-acceptance of e-learning platforms on one

hand and on the relationship between TAM constructs (PEOU and PU) and other external factors on the other hand.

## 7. Practical Implication

Fear of vaccination is a huge challenge that affects students' acceptance of technology. Hence, practical implications may help in leading the approaches and strategies in the learning environment. Researchers point out clearly that fear of vaccination may affect negatively users' perceptions. Accordingly, education practitioners and teachers should pay attention to students' perceptions and expectations during the e-learning models [37,130]. Accordingly, this paper has come up with certain practical implications. First, educators and health system workers should find different ways to increase the level of vaccination confidence so as to be accepted by users of technology. Second, researchers can get practical evidence of the effect that fear of vaccination may have bad consequences on the learning process by trying to investigate this issue further. In addition, the present study offers a scientific standpoint. Researchers in private and public institutions should consider the fear of vaccination as a crucial variable in the online learning environment, especially in the teaching and learning environment. Hence, teachers should reconsider the need to evaluate their instructional methods in teaching to meet the new challenges [131].

### 7.1. Managerial Implication

This research investigates the effect of vaccination hesitancy in the educational environment with reference for health care managers and governments to give priority to fear of vaccination and put forward effective steps to get rid of this fear. It gives a deep insight into current practices to reduce the level of fear and support the vaccination confidence among educators, teachers and students who will, in turn, affect society as a whole. For instance, healthcare managers should take into their consideration these findings to reduce the risk of the crisis that could result in vaccination rejection among students [132]. This research could enhance educational institutional teaching and learning efficiency and may lead to higher acceptance of vaccination which in turn enable the educational institutions to implement their goals and strategies more effectively.

### 7.2. Limitations of the Study

The study has some limitations. The main limitation is the participation of only three universities in the UAE, which does not allow full exploration of the factors affecting e-learning platforms after the spread of COVID-19. The study could have become more applicable with the participation of a greater number of universities. Further research will allow accurate comprehension of a mobile-learning system by fully investigating the factors, which affect e-learning systems. One limitation is the limited number of respondents in the research (545 students). The survey questionnaire was used as a means of data collection as per [84]. The research could have been better if a better instrument was employed with a better sampling technique. Moreover, the involvement of numerous universities from KSA, Kuwait and Bahrain from the Arab Gulf region would have improved research outcomes. It is essential to approach more students in future for taking part in such a study. Additionally, interviews and focus groups can prove better for more accurate outcomes. In the end, we look forward to the participating Arab universities implementing an e-learning system.

**Author Contributions:** Conceptualization, R.S.A.-M. and S.S.; methodology, I.A.; software, K.A.; validation, K.A., I.A. and S.S.; formal analysis, R.S.A.-M.; investigation, S.S.; resources, K.A.; data curation, I.A.; writing—original draft preparation, R.S.A.-M.; writing—review and editing, S.S.; visualization, K.A.; supervision, I.A.; project administration, R.S.A.-M.; funding acquisition, I.A. All authors have read and agreed to the published version of the manuscript.

**Funding:** This research received no external funding.

**Institutional Review Board Statement:** Not applicable.

**Informed Consent Statement:** Not applicable.

**Data Availability Statement:** Data developed in this study will be made available on request to the corresponding authors. The data are not publicly available because it is private dataset, Data is subject to corresponding author approval.

**Conflicts of Interest:** The authors declare no conflict of interest.

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
