# Peer review of "Factors That Affect E-Learning Platforms after the Spread of COVID-19: Post Acceptance Study"

_data, 2021_

Round 1

Reviewer 1 Report

The authors present a highly topical article, the review of the sources has a high thematic topicality index since about 50% of the references are from the last 5 years (2016-2021).

The structure of the article is adequate, the material and the method correctly describe the methodological procedures developed. The results are presented by means of a procedure based on a system of structural equations, Structural Equation Modeling (SEM), which is correct, and the discussion-conclusions are pertinent.

The purposes and hypotheses are coherent and the results related to them are adequately reported.

However, some changes are suggested, which are minor changes, for the improvement of the manuscript, after which it can be definitively admitted.

Accept after minor revision

ABSTRACT
It is necessary to readapt the abstract to the format: Introduction, Material-Method, Results and Discussion. No mention is made of the participants or the number of subjects in the sample in the abstract (n = 545). In addition, it is necessary to briefly inform the context where the study is carried out. Review the abstract format.

MATERIAL AND METHOD
Instruments
Although the reliability of each instrument is reported, Conbrach's alpha, it is advisable to use some other way, so it would be extremely useful for the authors to report the reliability of the scores through the Omega coefficient.

DISCUSSION-CONCLUSIONS
It is suggested that the number of citations / references in the discussion be increased, using more works from the last 5 years.

REFERENCES
The thematic news index is adequate, since about 50% of the references are from recent years.

Author Response

The authors are really very grateful to the feedback and comments raised by the reviewer which really assist them to significantly enhance this work and its presentation. The productive and valuable remarks enable us to update many parts of the paper as shown by the responses to each comment. Our responses are mentioned below under each comment raised by the reviewer and it is written in (Times New Roman, red color). Besides, all the updated parts in the manuscript were highlighted in yellow color in order to be easily tracked by the reviewers.

Reviewer 2 Report

Dear Authors, 

Thank you for presenting this interesting topic.

Although your paper is well define and the result was interesting but the short period of data collection is not acceptable and is not reliable to make decision on it. For the next round please collect data for longer period and apply your method on them.  

Author Response

(The authors gave the same response as above.)

Round 2

Reviewer 2 Report

Dear Authors, 

Thanks for pointing out my concerns, now it make more sense.

Have a wonderful weekend 

This manuscript is a resubmission of an earlier submission. The following is a list of the peer review reports and author responses from that submission.